# Relation of Whole Blood Amino Acid and Acylcarnitine Metabolome to Age, Sex, BMI, Puberty, and Metabolic Markers in Children and Adolescents

**DOI:** 10.3390/metabo10040149

**Published:** 2020-04-10

**Authors:** Josephin Hirschel, Mandy Vogel, Ronny Baber, Antje Garten, Carl Beuchel, Yvonne Dietz, Julia Dittrich, Antje Körner, Wieland Kiess, Uta Ceglarek

**Affiliations:** 1LIFE Leipzig Research Center for Civilization Diseases, University of Leipzig, Philipp-Rosenthal-Strasse 27, 04103 Leipzig, Germany; Josephin.Hirschel@medizin.uni-leipzig.de (J.H.); Mandy.Vogel@medizin.uni-leipzig.de (M.V.); Ronny.Baber@medizin.uni-leipzig.de (R.B.); Yvonne.Dietz@medizin.uni-leipzig.de (Y.D.); Antje.Koerner@medizin.uni-leipzig.de (A.K.); Wieland.Kiess@medizin.uni-leipzig.de (W.K.); 2Hospital for Children and Adolescents and Center for Pediatric Research (CPL), University of Leipzig, Liebigstrasse 20a, 04103 Leipzig, Germany; Antje.Garten@medizin.uni-leipzig.de; 3Institute of Laboratory Medicine, Clinical Chemistry and Molecular Diagnostics (ILM), University Hospital Leipzig, Paul-List Str.13/15, 04103 Leipzig, Germany; Julia.Dittrich@medizin.uni-leipzig.de; 4Institute of Metabolism and Systems Research, University of Birmingham, Birmingham B15 2TT, UK; 5Institute for Medical Informatics, Statistics and Epidemiology (IMISE), University of Leipzig, Härtelstrasse 16-18, 04107 Leipzig, Germany; cfbeuchel@imise.uni-leipzig.de

**Keywords:** pediatrics, amino acids, acylcarnitines, metabolomics, dried blood, tandem mass spectrometry

## Abstract

Background: Changes in the metabolic fingerprint of blood during child growth and development are a largely under-investigated area of research. The examination of such aspects requires a cohort of healthy children and adolescents who have been subjected to deep phenotyping, including collection of biospecimens for metabolomic analysis. The present study considered whether amino acid (AA) and acylcarnitine (AC) concentrations are associated with age, sex, body mass index (BMI), and puberty during childhood and adolescence. It also investigated whether there are associations between amino acids (AAs) and acylcarnitines (ACs) and laboratory parameters of glucose and lipid metabolism, as well as liver, kidney, and thyroid parameters. Methods: A total of 3989 dried whole blood samples collected from 2191 healthy participants, aged 3 months to 18 years, from the LIFE Child cohort (Leipzig, Germany) were analyzed using liquid chromatography tandem mass spectrometry to detect levels of 23 AAs, 6 ACs, and free carnitine (C0). Age- and sex-related percentiles were estimated for each metabolite. In addition, correlations between laboratory parameters and levels of the selected AAs and ACs were calculated using hierarchical models. Results: Four different age-dependent profile types were identified for AAs and ACs. Investigating the association with puberty, we mainly identified peak metabolite levels at Tanner stages 2 to 3 in girls and stages 3 to 5 in boys. Significant correlations were observed between BMI standard deviation score (BMI-SDS) and certain metabolites, among them, branched-chain (leucine/isoleucine, valine) and aromatic (phenylalanine, tyrosine) amino acids. Most of the metabolites correlated significantly with absolute concentrations of glucose, glycated hemoglobin (HbA1c), triglycerides, cystatin C (CysC), and creatinine. After age adjustment, significant correlations were observed between most metabolites and CysC, as well as HbA1c. Conclusions: During childhood, several AA and AC levels are related to age, sex, BMI, and puberty. Moreover, our data verified known associations but also revealed new correlations between AAs/ACs and specific key markers of metabolic function.

## 1. Introduction

Research into small-molecule metabolic intermediates in biological systems has become increasingly important in recent years as it offers insight into the interaction and regulation of human metabolic pathways. Disease-associated changes to metabolic pathways in human body fluids are examined to better understand the pathophysiology of complex diseases [1,2]. Various metabolic processes such as fatty acid oxidation, the carnitine shuttle system, the urea cycle, and the metabolism of branched-chain amino acids (BCAAs) can be characterized by measuring amino acid (AA) and acylcarnitine (AC) concentrations. In newborn screening, concentrations of amino acids (AAs) and acylcarnitines (ACs) are measured in dried blood spots (DBS) using tandem mass spectrometry to diagnose a variety of congenital metabolic disorders such as organic acidemia, aminoacidopathies, fatty acid oxidation disorders, carnitine cycle disorders, and some urea cycle disorders [3,4]. Previous studies have also investigated whether AAs and ACs can function as predictors of complex metabolic diseases in adults (e.g., type 2 diabetes, cardiovascular disease, obesity) [5,6,7,8]. More recently, metabolomics has also been applied in pediatric cohorts to identify biomarkers for predicting metabolic imbalances (e.g., early insulin resistance) or diseases [2,9,10]. Nevertheless, there are few pediatric epidemiological studies looking at physiological AA and AC concentration profiles and physiological processes and interactions associated with AA and AC metabolism. Possible explanations for this lack of studies include the requirement for recruiting a sufficient number of subjects of each age during childhood and the need to conduct a suitably thorough health assessment of each participant as well as ensuring standardized sample collection, measurement, and biobanking. The present study investigated AA and AC metabolome levels in whole blood in more than 2000 deep-phenotyped healthy children and adolescents from the LIFE Child cohort and related these levels to physiological changes during child development. In addition, associations with BMI, pubertal stages, and parameters of carbohydrate, fat, liver, kidney, and thyroid metabolism were also examined. 

## 2. Results

Laboratory parameters of carbohydrate, fat, liver, kidney, and thyroid metabolism of participants in the LIFE Child cohort that were included in this study are presented in Table 1. The individual number of samples for each listed biochemical parameter varies depending on the availability of sample material (total: *n* = 2903–3711 samples; females: *n* = 1445–1828 samples; males: *n* = 1458–1883 samples).

### 2.1. Influence of Age and Sex on AA and AC Levels

Our first objective was to describe normal alterations of AAs and ACs during the first 18 years of life, including sex differences. Age-dependent variations in the concentration levels of 23 AAs, 6 ACs, and free carnitine were expressed as 2.5th, 10th, 50th, 90th, and 97.5th percentile curves (Appendix A). Appendix A provides the numerical values of the percentiles of each metabolite for females and males from 0.25 to 18 years of age. In addition, the parameters of location, scale, and skewness are shown. Even though each metabolite has a unique concentration profile, we observed that the age-dependent changes in the analyzed metabolites could be categorized into four main profile types. An example of each profile type is presented in Figure 1a–d. Significance analyses of the individual metabolite profiles are shown in Appendix A.

The first profile type is characterized by higher concentrations during early infancy (3 months to 1 year of age), followed by a decrease until the age of 1–5 years. Thereafter, concentration levels remain approximately stable or increase up to the age of 18 (Figure 1a and Appendix A). This ‘type’ of age-dependent profile was seen for alanine (Ala), arginine (Arg), glutamic acid (Glu), leucine/isoleucine (Leu/Ile), ornithine (Orn), proline (Pro), serine (Ser), threonine (Thr), tyrosine (Tyr), free carnitine (C0), acetylcarnitine (C2), hexadecanoylcarnitine (C16), and octadecenoylcarnitine (C18:1). Leu/Ile, Tyr, C0, and C18:1 differed in females, with slightly decreasing concentrations during preadolescence or adolescence. For Ala, Glu, Pro, Ser, Thr, C2, and C18:1, the observed decrease in early childhood and the following increase were both found to be significant (*p* < 0.001 to *p* = 0.022) in females and males (Appendix A). Significant sex differences were found in at least one age interval for Arg, Glu, Leu/Ile, Orn, Pro, Ser, Tyr, C0, C2, C16, and C18:1.

The second profile type features a continuous increase from the age of 3 months to 18 years (Figure 1b and Appendix A). This type includes aminobutyric acid (Aba), aspartic acid (Asp), glycine (Gly), histidine (His), hydroxyproline (OH-Prol), methylhistidine (MeHis), pipecolic acid (PiPA), phenylalanine (Phe), tryptophan (Trp), and valine (Val). The increase in mean concentration with increasing age was significant for all metabolites (*p* < 0.001) except for Phe in girls and Trp (Appendix A). The level of Trp increased significantly in the age range from 4 to 10 years. For Val, this profile was only observed in boys, while for Aba, Gly, and OH-Prol, continuous increases were only observed in girls up to the age of 11, 13, and 12 years, respectively. Significant sex differences were found in Aba, Gly, OH-Prol, MeHis, and Val levels.

The third profile type includes those of citrulline (Cit) and sarcosine (Sarc), with mean concentrations increasing significantly (*p* < 0.001) until the ages of 9 and 5 years, respectively (Figure 1c and Appendix A). This was followed by stable values until the age of 18, except in the case of Cit in girls, which exhibits a decreasing trend after 9 years of age. No significant sex differences were found for either of these two metabolites (Appendix A).

The fourth profile type, which applies to methionine (Met), taurine (Tau), propionylcarnitine (C3), octadecanoylcarnitine (C18), and methylmalonylcarnitine (MMA), does not exhibit any sort of trend in boys or girls (Figure 1d and Appendix A).

### 2.2. Associations of AA/AC Levels with BMI and Puberty

Next, we investigated whether AA and AC levels are related to obesity or pubertal status. The distribution of pubertal stages and BMI-SDS are presented in Table 2 and Table 3. 

The results of the linear regression analysis for each metabolite with BMI (SDS values) and Tanner stage (absolute values) are provided in Appendix A. Metabolites that exhibited significant sex differences with regard to associations with BMI-SDS (p_adj_ < 0.05) have been presented separately for females and males, concerning Gly, Sarc, C2, and C3. Significant positive associations with BMI-SDS were found for Leu/Ile, Tyr, Val, C0 (p_adj_ < 0.001), Ala, Phe, Pro (p_adj_ < 0.01), and OH-Prol (p_adj_ < 0.05) in both girls and boys. The relationships between BMI―expressed as BMI-SDS―and the mean SDS levels of Leu/Ile and Val are shown in Figure 2 by way of example. In addition, we detected significant positive correlations with BMI-SDS for C3 (p_adj_ < 0.001), Sarc (p_adj_ < 0.01), and C2 (p_adj_ < 0.05) in males. Significant negative associations with BMI-SDS were found for Gly (p_adj_ < 0.01) in males and for Cit (p_adj_ < 0.05) in both sexes. 

The comparison of the pubertal and post-pubertal stages with the pre-pubertal stage (Tanner stage 1 as the reference level) showed significant (p_adj_ < 0.05) differences in metabolite concentrations in at least one Tanner stage (with the exception of Glu, Met, Orn, Phe, C2, C3, C16, C18, and MMA). For Asp, Gly, MeHis, Thr, and Trp, the concentrations in all Tanner stages differed significantly from Tanner stage 1 in both girls and boys. Significant (p_adj_ < 0.05) sex-dependent differences were found in Tanner stages 4 and/or 5 (Appendix A). Significant sex differences exhibited by BCAAs in Tanner stage 4 and/or 5 are presented by way of example in Figure 3. 

### 2.3. Correlations between AA and AC Levels and Metabolic Markers

Last of all, we checked whether biochemical markers for carbohydrate and fat metabolism and liver, kidney, and thyroid function were associated with changes in AA and AC levels. The heatmaps, presented in Figure 4, summarize the correlations between the metabolites and the biochemical parameters based on absolute (Figure 4a) and SDS-normalized concentrations (Figure 4b). The correlation coefficients and p-values are provided in Appendix A. Five clusters were identified in the different correlation patterns. These clusters were mainly defined by the correlations of the metabolites with CysC, creatinine, triglycerides, glucose, and HbA1c. For HbA1c and CysC, these correlations remained highly significant after age-adjustment (p_adj_ < 0.00001 to p_adj_ < 0.05). The majority of metabolites were significantly positively associated with blood glucose level but significantly negatively correlated with HbA1c, even after age adjustment. Furthermore, most of the metabolites showed significant positive associations with CysC. SDS-normalized triglyceride levels were significantly positively associated with metabolites in Cluster 5, namely Ala, Arg, Pro, Thr, and Tyr as well as with Aba, Cit, Gly, Leu/Ile, MeHis, Met, Sarc, and Val (p_adj_ < 0.00001 to p_adj_ < 0.05). By contrast, total cholesterol, LDL-Chol, HDL-Chol, ALT, AST, TSH, FT3, and FT4 showed only a few weak correlations in any of the five clusters.

## 3. Discussion

This study analyzed AA and AC concentrations in dried blood spots (DBS) in a healthy pediatric population ranging from 3 months to 18 years of age (see Table 1). Whereas numerous studies have been done in the field of newborn screening looking at dried blood levels of AAs and ACs, data regarding age- and sex-dependent variations of AA and AC concentration levels during child development in a healthy population are hard to come by. To our knowledge, only a few studies conducted to date have focused on AA and AC changes in serum, plasma, or dried whole blood during childhood [11,12,13]. These studies were mostly based on a smaller number of subjects (148–500 children) and did not consider associations with laboratory parameters relating to carbohydrate, fat, liver, kidney, and thyroid metabolism. Moreover, the analyses of correlations or confounding factors were limited to age and did not consider pubertal state and BMI. The LIFE Child study follows highly standardized procedures in terms of subject instructions and recruitment with medical examinations at the study site always starting with blood sample collection, followed by the participant having breakfast [14]. As indicated by the concentration values for triglycerides and glucose in Table 1, the required overnight-fasting conditions were mostly complied with, making this cohort specifically appropriate for the analyses conducted in this study.

With the exception of Met and Tau, all AA concentrations changed between the ages of 3 months and 18 years, though they followed different patterns. For carnitines, we found age-dependent changes in concentration levels for C0, C2, C16, and C18:1. Four different profile types were identified, as shown in Figure 1a–d. For example, the concentration values for AAs grouped in profile 1 decreased during the first years of life, with most tending to increase continuously thereafter. Similar results have been shown in previous studies [11,13]. However, the groupings of AAs and ACs defined by these profiles could not be assigned to specific metabolic pathways. In addition, no similarities were found in regulation patterns regarding age-dependent changes in proteinogenic/non-proteinogenic and essential/non-essential AAs. 

Our results indicate a strong association between BMI-SDS and certain AA and AC values in a healthy cohort. BMI and levels of both branched-chain and aromatic AAs were significantly positively associated in both sexes. These findings were also demonstrated in previous studies focusing on obesity in adults [15,16] and individuals with type 2 diabetes [6,8,17] as well as in studies focusing on obesity in children [18,19]. McCormack et al. and Perng et al. also indicated that elevated levels of BCAAs are significantly positively associated with obesity in children and adolescents [18,19]. In addition, similar to Perng et al., our results showed that C3 carnitine―as an intermediate of the BCAA metabolism―also revealed a highly significant positive association with the BMI in males [19]. Significant negative relations between metabolite concentrations and BMI were only found for the two metabolites Cit and Gly (only in females). The negative relationship between glycine and obesity was also reported by previous studies [16,20,21]. The results of the present study show that there is an association between weight status (BMI) and the AA/AC metabolism even under physiological conditions (based on a cohort of non-diabetic subjects), and thus suggest that amino acid and acylcarnitine levels could be applied as biomarkers for metabolic alterations or disorders in overweight/obese individuals. Apart from these findings, our results revealed that puberty has a major influence on AA and AC metabolism: by way of example, the concentration profile for BCAAs presented predominantly increasing values in boys (Tanner stage 2–5) and decreasing values after Tanner stage 3 in girls (Figure 3).

Our study also aimed to assess possible correlations between AAs and ACs and key markers of metabolic function. Surprisingly, we found that kidney function (assessed by CysC and creatinine) was observed to have a substantial impact on cluster formation with mostly significant positive associations. For CysC, this effect was still present after adjustment for age. To our best knowledge, only a few studies are available that examine circulating levels of CysC under physiological conditions in children. Age-dependent reference ranges up to the age of 18 have been published by Ziegelasch et al. [22]. In addition to being a marker for kidney function, CysC is involved in several pathophysiological processes. CysC is a potent extracellular cysteine protease inhibitor that regulates, for example, protein turnover and pro-protein processing [23]. It is still unclear if the observed correlation between CysC and the metabolites can be explained by kidney function or other biological relationships reflected by CysC. 

Moreover, our results demonstrate that most of the investigated metabolites showed a significant positive association with glucose, but a highly significant negative correlation with HbA1c, even after age adjustment. This effect is surprising because in adults, the inverse effect is described in numerous studies, mostly in the context of type 2 diabetes [24]. To date, only limited information on this association is available for children [10,25,26]. Perng et al. described a positive correlation between BCAAs and glucose and a negative association between BCAAs and C-peptide insulin resistance (CP-IR) in a smaller study of 179 adolescents [10]. Mihalik et al., meanwhile, reported lower plasma concentrations of most AAs and short- and medium-chain ACs in obese adolescents with type 2 diabetes (with a higher HbA1c level). Their explanations for these differing findings between adolescents and adults were reduced catabolism or higher anabolism during development and growth in adolescents as well as an elevated rate of gluconeogenesis (increased use of glucogenic AAs as substrates) and beta-oxidation (increased use of ACs as intermediates) in diabetic subjects. As such, they hypothesized that younger people exhibited greater metabolic plasticity and adaptation [26]. Lamichhane et al. indicated a significant negative association between protein and leucine intake and HbA1c in young people with type 1 diabetes, which they concluded pointed to a beneficial effect of protein and leucine supplementation on HbA1c levels [25]. Interestingly, these results were confirmed in our study with a healthy, non-diabetic pediatric cohort and indicate that the inverse association between AAs/ACs and HbA1c can already be observed under physiological conditions in children and adolescents.

Triglycerides showed significant positive associations with ketogenic and glucogenic AAs based on both absolute and SDS-normalized values. However, significant correlations between triglycerides and metabolites of beta-oxidation and the carnitine cycle were only identified when using absolute concentration values. Similarly to our results, Wiklund et al. also found significant positive correlations between Ala, Leu, and Ile and triglyceride levels (in adolescent girls), and suggested that these AAs could predict hypertriglyceridemia in early adulthood [27]. Regarding laboratory parameters known to reflect cholesterol metabolism and thyroid and liver function, we could not identify any relevant association between these parameters and the measured metabolites. After age adjustment, only a few associations remained statistically significant, but these were not linked to specific metabolic pathways.

To the best of our knowledge, this is the first study of a large pediatric cohort in a Caucasian/German population investigating physiological AA and AC concentrations and associations with biochemical parameters. The strengths of our study include the number of metabolites investigated, i.e., 30 (23 AAs, 6 ACs, and free carnitine) and the large sample size (2191 subjects aged 3 months to 18 years; 3989 DBS samples). A further strength of the LIFE Child cohort is the availability of data regarding the participants’ puberty status and BMI, and concentrations of key metabolic serum markers. Furthermore, sample collection, pretreatment, biobanking, and comprehensive analysis followed highly standardized assessments and procedures. 

Regarding the measurement methodology, one limitation is the inability to discern leucine and isoleucine. Moreover, the use of age-grouped batches for MS/MS analysis meant that statistical batch adjustment could not be applied although quality controls were used for each batch to minimize batch effects [28]. Another limitation is the impossibility of ensuring that all participants adhered to the fasting requirement. Since all tests were conducted in the same cohort, independent replication in an external cohort would be required for validation of the findings. Finally, it should be acknowledged that from an initial 62 metabolites, only 30 met the requirements for further procedures and analyses. At the same time, however, this can be seen to reflect the high quality of control standards applied in this study.

## 4. Materials and Methods

### 4.1. Study Population and Design

LIFE Child is a prospective, longitudinal, population-based childhood cohort study carried out at the Leipzig Research Center for Civilization Diseases (LIFE) in the city of Leipzig (Saxony, Germany). Participants join the study at any age from 3 months to 16 years, and are asked to attend annual follow-up visits until the age of 20. For participants recruited during the first year of life, data were collected at 3, 6, and 12 months of age. The majority of participants are Caucasians who live in Leipzig and the surrounding area. The LIFE Child study was designed to investigate the impact of metabolic, genetic, and environmental factors on health and development in children and adolescents. It was approved by the Ethical Committee of the University of Leipzig (reference number: Reg. No. 264-10-19042010) and is registered at ClinicalTrials.gov (NCT02550236). All procedures are performed in accordance with the ethical standards of the institutional research committee and the Declaration of Helsinki [14,29,30]. All parents and participants aged 12 years or older gave informed written consent for each study visit. The data of all individuals are pseudonymized, as has been described elsewhere [14]. The sub-study described here included 3989 measurements from 2191 children aged between 3 months and 18 years (Scheme 1 and Appendix A). Children aged 4 years and over were instructed to fast overnight for at least 8 h. The data collection for this study was carried out between May 2011 and December 2014.

### 4.2. Sample Pretreatment and Analysis

EDTA whole blood was obtained by venous blood sampling (EDTA monovettes: Sarstedt AG & Co, Nümbrecht, Germany) and the samples were processed by the LIFE preanalytical laboratory following standardized protocols. EDTA whole blood was spotted on grade 903 filter paper (Whatman GmbH, Dassel, Germany) using five 40 μL portions within 1 h after blood withdrawal. For younger children (primarily from 3 months to 2 years in age), whole blood was directly dispensed onto the filter paper during open blood collection in the case of a small quantity of sample material. Samples were dried for 3 to 24 h at room temperature then stored, until pretreatment, at −80 °C in Foil-Barrier Resealable Bags containing MiniPax absorbent packets (Whatman GmbH, Dassel, Germany). The sample pretreatment and analysis protocol was applied according to previously described, validated procedures [28,31]. Briefly, 3 mm diameter dried blood spots (containing 3 µL of whole blood) were punched out using a Multipuncher (PerkinElmer Wallac GmbH; Freiburg, Germany), then extracted and derivatized with butanolic HCl in a 96-well microtiter plate with methanol-containing isotope-labeled standards. Subsequently, flow injection analysis (FIA) was performed using an Atmospheric Pressure Ionization (API) tandem mass spectrometer (API 2000, Sciex, Darmstadt, Germany). Tandem mass spectrometry (MS/MS) analysis was performed on 60 age-group batches, with each plate including two commercially available quality control samples from Recipe (Munich, Germany) for estimation of precision and accuracy (Appendix A). Concentrations of 26 AAs, 34 ACs, and free carnitine were quantified using ChemoView™ 1.4.2 software (Sciex, Darmstadt, Germany). The accuracy, linearity, and method validations are described in Brauer et al. [28]. According to Brauer et al. [28], which listed recovery rates for each AA, recovery rates were calculated for each AA detected in the present study (Appendix A). Additionally, either enzymatic or immunoassay analysis, carried out using a COBAS 6000/8000 lab analyzer system (Roche, Mannheim, Germany), was used to measure concentrations of the following substances: glucose and HbA1c; triglycerides, total cholesterol, low-density lipoprotein cholesterol (LDL-C), and high-density lipoprotein cholesterol (HDL-C); alanine aminotransferase (ALT) and aspartate aminotransferase (AST); cystatin C (CysC) and creatinine; thyroid-stimulating hormone (TSH), free tri-iodothyronine (FT3), and free thyroxine (FT4) [28].

### 4.3. Statistical Analysis

The analytical plan, including data pre-processing, data analysis, and the respective output, is shown in Appendix A. Metabolites with >5% of concentrations below the detection limit were excluded. Glutamine and lysine could not be included in the analysis due to perturbing batch effects probably caused by the lack of reproducible butyl ester formation during derivatization. From an initial 62 metabolites, 30 passed quality control criteria for further analysis (Appendix A). Furthermore, to stabilize the analyses, outliers were removed by applying a cutoff of ±5 standard deviations based on the mean of the logarithmized nonzero data [32,33].

Trends in metabolite concentrations were regressed continuously on age for each sex separately using generalized additive models for location, shape, and scale, as implemented in the R package “gamlss” [34]. The generalized additive model for location, scale, and shape models not only the mean value but also scale and shape parameters of the age-varying distribution of the response variable [34]. The age- and sex-dependent distributions were estimated applying a modified LMS (L: skewness; M: mean; S: coefficient of variation) method, assuming a Box–Cox t (BCT) distribution for aspartic acid (Asp), methionine (Met), and taurine (Tau) and a Box–Cox Cole and Green (BCCG) distribution for the remaining target variables. The BCT distribution (a four-parameter distribution) was only applied if data of the respective metabolite required an additional adjustment for kurtosis. Otherwise, the three-parameter distribution (location, variation, skewness) was preferred according to the statistical principle of parsimony. Models were compared using the Akaike and Bayesian information criterion [35,36,37]. To adjust for multiple measurements for a single family (siblings were included) or a single subject, the LMS method was combined with resampling, as has been described in detail elsewhere [37]. Quantile–quantile plots (QQ-plots) were applied to assess model quality [38]. Metabolite concentration levels show a nonlinear trend over age. However, linearity can only be assumed within age intervals. Therefore, hierarchical multiple linear regression was applied to the data based on age intervals. A detailed description of the stepwise approximation can be found in Appendix A.

To assess possible correlations between each metabolite and the biochemical parameters, hierarchical models were applied for both absolute and SDS values, with the results visualized using correlation matrices. Hierarchical modeling was applied to adjust for multiple measurements per subject. Associations between metabolites and BMI-SDS or pubertal development were also estimated using hierarchical multiple linear regression. To consider the weight status (BMI) in dependence from age and sex, BMI values were converted to standard deviation scores (SDS) using the German age- and sex-specific norms [39]. Moreover, the interaction of sex and BMI-SDS were tested and results were reported for each sex separately in the case of significance (p_adj_ < 0.05). The association with puberty was analyzed as stratified by sex. Age- and sex-adjusted SDS values for all AAs and ACs, for creatinine, glucose, and for HbA1c were calculated using the sds-function “childsds” in the R-package [37,40]. The relative references for triglycerides, total cholesterol, LDL-C, HDL-C, AST, ALT, and CysC have been previously published [22,41,42]. References for the thyroid hormones (TSH, FT3, and FT4) are still unpublished and were also estimated using the LMS method. 

To carry out hierarchical modeling, functions as implemented in the R package “lme4” were applied and standardized coefficients were extracted [43]. To estimate the respective p-values, the R package “multcomp” was used [44]. The *p*-values were adjusted for multiple testing (p_adj_) by applying the method of Benjamini, Hochberg, and Yekutieli for controlling the false discovery rate (FDR = 5%) [45]. Data analyses were carried out using R software (version 3.6.1; R Foundation for Statistical Computing, Vienna, Austria) [46]. The statistical significance level was set to *α* = 0.05.

## 5. Conclusions

Our study indicates that most of the investigated metabolites exhibit age-dependent concentration variations and that some of them differ by sex. In addition, our results demonstrate an association of BMI and puberty with AA and AC levels as well as the strongest correlations of AAs/ACs with carbohydrate, fat, and kidney metabolism, which most likely indicate interacting metabolic pathways. This study describes age- and sex-related percentiles for AAs and ACs as well as associations of these metabolites with various biochemical parameters and is, therefore, a useful reference for future studies in the field of metabolism and biochemistry. Further studies are needed to continue mapping and identifying relevant pathophysiological metabolic pathways with the objective to potentially apply AAs and ACs as early biomarkers to identify individuals at increased risk of developing metabolic diseases.

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
