# Peer review of "Relation of Whole Blood Amino Acid and Acylcarnitine Metabolome to Age, Sex, BMI, Puberty, and Metabolic Markers in Children and Adolescents"

_metabolites, 2020, doi:10.3390/metabo10040149_

Round 1

Reviewer 1 Report

Authors have provided enough explanations for my concerns about this papers.

Reviewer 2 Report

Thank you for addressing my comments in a thorough and respectful manner, and for clarifying instances where I simply misunderstood or was unaware of common practices. I am satisfied with all the author comments/corrections and have commented on them individually below.

Comment:

Comment 1: Supplementary table 4 is very useful and well organized. It summarizes the data very well.

Comment 2: A valid point. The section is fine as is, however, in the event that a reader has a similar thought, is it possible to indicate that you were using a method validated specifically for these metabolites?

Comment 3: Fair point and this is clear in the paper.

Comment 4: Thank you for the clarification. This was misunderstanding on the part of the reviewer.

Comment 5: I really like the addition of Table S5. It will appeal to those in analytical chemistry and metabolomics and contains a lot of useful methodological information.

Comment 6: Noted as above.

Comment 7-9: The information of the 2 level commercial QC is extremely useful and addresses my questions. Furthermore, the inclusion of Supplemental Scheme 1 is excellent. It clearly illustrates the entire study from a data perspective and should be required for all studies! I also like the PCA plot and could see if being useful to include as a supplemental figure, though it isn’t necessary.           

Comment 10: The conclusion is concise and summarizes the results well. It also acknowledges the work done in the current study to create a valuable reference and foundation for future studies. It indicates clear directions forward to further develop the field.

Comment 11: I understand and appreciate that outside of a validated IVD kit, results from one mass spec vendor and/or model are not directly applicable to another. However, reference ranges can be useful, even if simply to ensure results from other studies are in the same order of magnitude. Table S1 is a useful tool, particularly given the age range covered.

Comment 12: Very helpful.

Comment 13: Thank you for the explanation. It is clear to me now.

Comment 14: Noted.

Comment 15: The change to the manuscript makes this clear.

Comment 16: The change makes the context of the study clear and accurate.

Comment 17: This change clarifies the exclusion of some key metabolites.

Comment 18: Clear and noted.

Comment 19: This was addressed in the updated conclusion section.

This manuscript is a resubmission of an earlier submission. The following is a list of the peer review reports and author responses from that submission.

Round 1

Reviewer 1 Report

Review Metabolites: “Relation of Whole Blood Amino Acid and 3 Acylcarnitine Metabolome to Age, Gender, BMI, 4 Puberty, and Metabolic Markers in Children and 5 Adolescents”.

General comments:

The Authors address a relevant subject in their evaluation of whether factors like age, sex and obesity, as well as several other laboratory measurements, were related to concentration levels of amino acids and acylcarnitines in children. The study makes use of very unique resources from the LIFE Child cohort (Leipzig, Germany) with metabolomics data from 3,989 dried whole blood samples collected from 2,191 healthy participants.

However, in its current form, the study and the manuscript have three important limitations:

First, a coherent analytical plan is lacking. The role of BMI, sex and age are evaluated in separate association analyses that do not account for the large inter-correlations among these factors. It is very difficult to judge the validity of findings in the absence of methodology that handles this complexity. In this regard, it is not obvious to capture the choice of displaying some of the results presented in the manuscript rather than others. For example in Figures 2 and 3, relationships are shown “by way of example”, but the scientific interest and merit to have them displayed is not justified. There is no mention of methods to control the nominal level of statistical significance for multiple comparisons, given the large number of tests performed, with the exception of the ‘multcomp’ command in line 360. Readers expect an exhaustive description of the methodology used rather than suboptimal references to R commands.

Second, although longitudinal measurements were available, it is hard to judge whether this feature was fully exploited during the analysis. Hierarchical models with subjects-specific terms as random-effects should be consistently used in the analysis. The current description of statistical analysis does not clarify whether this was the chosen analytical strategy. In addition, the jargon in the manuscript often uses causal statements, which are inappropriate given the observational nature of the study, particularly because the longitudinal nature of the study does not seem fully exploited. Terms like ‘effects’ (line 145) or ‘influence’ (lines 40, 352) are used to describe associations, and should therefore be avoided, and a more prudent language should rather be employed throughout. One simple suggestion would be to use less but more targeted statistical techniques, and explain them more clearly.

Third, as a consequence of the previous points, it is very difficult to understand the logic of the Result section as a whole. The choice of reporting some of the findings does not follow clearly defined criteria, and lists of metabolite/variable (BMI, age, sex) combinations seem chosen from a large basket of potentially relevant results. Last, as detailed below, some important details are given for grant, while their use and meaning should be more clearly explained. For example, what does LMS in the Abstract stand for?

Detailed comments:

Lines 126: the role of obesity, age and sex should be investigated simultaneously, thus controlling for mutual potential confounding, rather than separately, as the text seems to suggest.

Distributions of Tanner stage values by sex in Table 2 indicate that males were overall younger than females, a result that calls for a careful definition of the analytical framework to appreciate important links;

Line 138; what does the p-value indicate, possibly an interaction between sex and interaction? There is no mention of use of interaction terms in the statistical analysis section.

Lines 149-150: the text reads very obscure, please consider revision.

Line 332: exclusion of outliers using deviations from the mean should be done after transformation of likely skewed distributions; otherwise the choice of this method is suboptimal for outlier identification and exclusion;

Line 334: The expression “…were estimated continuously” is improvable, for example as: “Metabolite concentrations were regressed on age in continuous using generalized additive models … “; also, the text would be more explicit if details on the generalized additive models for location, shape and scale were provided;

Line 336-339: how was the choice of t- and Cole and Green distributions determined for the different sets of metabolites? Lines 341-343: the text is very obscure, please consider revision. What intervals and cut-off points do the Authors refer to?

Lines 344-346: The description of the methodology is very confusing here. The choice of multiple linear modeling is a good one (a mixed model would be even better), but why is age modeled through age-interval and in continuous? And why BMI is not included as covariate as well? What about age and sex interactions to appreciate heterogeneity of associations by gender?

Lines 347-348: Similarly, the use of hierarchical models is a good choice, and they are instrumental to evaluate potential relationships rather than assessing correlations. What does SDS stand for?

Figure 4: it is frankly very difficult to appreciate the correlation values reported in the heatmaps, and even more so what the groups indicate. Please consider profound revision. In other terms, what laboratory parameters are reported horizontally? The expression ‘laboratory parameters’ is improvable, please consider revision.

Reviewer 2 Report

In this study, the authors aimed to examine the relation of whole blood amino acid and acylcarnitine metabolome to age, gender, BMI, puberty, and metabolic markers in children and adolescents.

The work of this paper is interesting and provides the important information for metabolic function in children and adolescents. There are some minor comments.

Discussion section

P8, lines 211-224: should indicate physiological meanings from results of present study.

Materials and methods

P11, lines 330-333: should specify the excluding and including metabolites in table or supplemental data.

P11, lines 356-357: should not include this sentence as this information is not published data.

I think that English should be improved as academic writing. 

Reviewer 3 Report

Overall, this is a valuable resource to those in the fields of metabolomics, metabolics and child health. Publishing reference ranges across ages and genders for a larger-than-usual sample size is a benefit to the broader scientific community. It is well written and easy to follow. The language is appropriately technical and the arguments made for why the study was undertaken and valid and persuasive: there is a lack of data on AA and AC levels beyond the neonatal stage and certainly little known on their correlations to key developmental stages and other physiological/biochemical parameters.

A few comments to address are as follows. I have broken them down by paper heading:

Introduction section:

Is there another reason why the analysis was limited to only AA and AC? Why not look into organic acids, fatty acids, lipids? Many inborn errors are not diagnosed solely based on AA and AC (line 54-55) but rely on more specific tests looking at other metabolites the relevant biochemical pathways.

Method section:

The methods section was lacking in some key details that made interpretation of the results somewhat challenging. While the study design and sample recruitment and sample filtering was suitable, some of the sample pretreatment and analysis was confusing or lacking. Lines 305-306: was the blood kept on ice prior to spotting to minimize metabolic changes? It might be prudent to mention this, particularly given that blood in younger children was collected slightly differently Lines 307-308: “…blood was directly dispensed onto the filter paper during open blood collection…” I am unclear as to the meaning of this (this could be my own comprehension and not the authors’ words). Does this mean it was still collected in a monovette and then dropped onto the card? Could the sentence be edited to read “…blood was directly dispensed from the monovette onto the filter paper during open blood collection…” Line 314: Was there a matching D-IS for every compound? Line 318: Were there any limitations to quantitation because of the IS? Line 321: What was the recovery for each AC? Why was it not reported? It might be helpful to report CVs for each included compound as well as recovery. Is there a reason that the samples were analyzed in batches by age and not randomly? I know the limitation was also addressed in the results section (line 273) but it might be nice to clarify why it was done (i.e. based on receipt and recruitment, etc.) Lines 313-314: Though the protocol is described in other papers, the specific derivatization chemistry (i.e. butanol in HCl) should be specified to ensure that the results can be interpreted accurately. Line 317: Could you clarify the QC material used? Were these a high/low QC, beginning and end of run, etc. Are these made from a pool of the samples themselves, or are these standardized, purchased materials? As an example to the above comment, glutamine can often be hydrolyzed to glutamic acid (lines 330-331) and this might a) explain why there were deviations in the QC and b) might impact true quant of glutamic acid (the same can apply to asparagine/aspartic acid). Furthermore, the potential isobaric interferences of glutamine/lysine might be contributing to the “perturbing batch effects” and subsequent removal of the metabolites from the study (line 331) Line 332: Could you elaborate on the QC procedures used to filter the data? Was this a QC CV < x%, linearity, LOD, etc.? If you quantified the same QC materials across each plate, why could batch correction not be applied to the data (as in reference 32)? Is there a figure which could illustrate these batch effects, such as a Levey-Jennings QC plot or PCA scores plot?

Conclusion section:

In the line conclusions section (lines 337-338) are the authors suggesting that further pathway mapping and identification is beyond the scope of the present work? I suspect this is the case, so why not explicitly call on the greater community to step up? Lines 369-370 do a good job of this. It might be worthwhile to consider also framing this work as a nice reference for future studies in metabolism and biochemistry. While other studies have published ranges for a smaller number of compounds or across a larger age range, AA and AC profiles broken down by year and gender are extremely useful.

Results section:

A table listing the concentrations (reference ranges) stratified by age and gender either in the body of the text or as a supplement would be useful as a reference for other studies and other researchers. This would be especially prudent as the authors, rightfully, claimed that few studies focus AA and AC in children. In table 1, is it possible to indicate in some way the organ or pathway each lab parameter relates to? While some are obvious (i.e. fat metabolism and triglycerides) liver and kidney function may be less obvious to some more chemistry- or analytical-based readers Lines 123-124. When stating there is no trend in boys and girls, does this mean collectively as a group or statistically within each metabolite? At least by eye, Met appears to decrease over time, but is this not sig?

Discussion section:

Line 232: When stating that there are other pathophys processes that implicated CysC, could you give a few examples? This might tie in with claims in line 235 that CysC correlation cannot be conclusively tied to kidney function but rather to other processes Line 253: can you clarify was “these effects” refers to? Are you claiming that you are seeing a reduction in HbA1c because of increased AA/Leu? If so, this seems like a bold claim to make, especially since Leu/Ile/OH-Pro are quantified together. If you mean metabolic plasticity seen in the Mikhail paper, simply clarify. Lines 264-265: Given that pathway analyses/mapping weren’t possible, I’m not sure if stating that this is the first study to investigate AA/AC and “corresponding metabolic pathways” is entirely accurate. It might be more prudent to say “biochemical parameters” or something of this nature. Lines 272-273: As mentioned previously, forming butyl esters can preclude the analysis of Asn and Gln and compromise Asp and Glu quantitation Lines 278-280: Previous comments in the methods section addressed this, but the QC procedures and batch effects were not clear. While I agree that stringent/conservative data filtering can provide high-quality data a bit more transparency to the reader can help convince people. Future directions for this work were not clear. For instance, expanding metabolome coverage (either by additional targeted approaches, Biocrates kits, additional instrumental platforms, untargeted approaches etc.) could help with pathway mapping and provide deeper insight into metabolite/biochemical associations.

Round 2

Reviewer 1 Report

The Authors addressed some of the suggestions received, but failed to address major elements of the analytical and interpretation process of their interesting work. A more rigorous analytical plan coupled with a more organized presentation of results and the implementation of procedures to account for multiple testing, as customarily done in statistical analyses of large numbers of features, seem necessary steps to produce accurate scientific conclusions.
